# The Resilience of Tourism Recreation Companies in a Pandemic Context: The Case of Canyoning in the Azores

**Francisco Silva** [1,*] **, Tiago Lopes** [1] **and Mário Silva** [2]

1   Centre of Geographical Studies, Instituto de Geografia e Ordenamento do Território (IGOT),
    University of Lisbon, 1600-276 Lisbon, Portugal
2   Centre for Tourism Research Development and Innovation (CiTUR), Escola Superior de Hotelaria e Turismo
    do Estoril, 2769-510 Estoril, Portugal
*   Correspondence: francisco.silva@eshte.pt

**Abstract:** The COVID-19 pandemic had an enormous impact on the tourism sector. Economies with a large weight of external tourism were especially affected, as was the case with the Azores region. This paper aims to analyse the level of this crisis and the resilience of tourism recreation companies specializing in canyoning in the Azores region, and whether these reactions led to more sustainable business models. The adopted methodology considered the analysis of recent statistical data on tourism and the available information on COVID-19 pandemic-related impacts. In parallel, questionnaires were applied to all tourism recreation companies in Portugal, and semi-structured interviews were conducted with all the canyoning providers in the Azores region. Results show that all companies survived the crisis, and some have made significant changes to their business models. Furthermore, some degree of success was observed in how services adapted to new demand characteristics, given the small size of most of these businesses and the specialization of their products.

**Keywords:** Azores; COVID-19; canyoning; entrepreneurship; resilience; tourism





## 1. Introduction

The COVID-19 pandemic has created an unprecedented crisis in the tourism sector, with an average 70.1% reduction in international tourism arrivals in 2020, and an economic contraction in Europe estimated at 67.7% (UNWTO 2022). In the European tourism sector, cruises and airlines have taken the most severe blow, reporting a plummet of 90%, followed by tour operators and travel agencies (70%), and hotels and restaurants (50%) (European Parliament 2020). Between late March and early April 2020, 156 world destinations imposed major restrictions on international travel, some of which even closed their borders, which resulted in a dramatic plunge in international tourist arrivals, with an approximate year-on-year reduction of 97% between April and May (UNWTO 2022).

The intensity of this pandemic and its regional distribution showed great variability, with several "waves" of COVID-19. In 2021, some countries managed to significantly reduce restrictions, especially in periods with fewer infection records, which allowed some tourism recovery (Okafor et al. 2022). However, many countries continued to impose limitations on international travel.

With widespread vaccination and the disease's tendency to transition from pandemic to endemic status, countries were able to lift most restrictions by 2022. Consequently, a quick recovery in travel and tourism began, with some destinations, such as Portugal, exceeding in summer 2022 the pre-pandemic results of 2019 (INE 2022). It is believed that this recent demand was driven by a "burning urge" to travel, showing important "psychological roots, signifying travellers' need for retribution against COVID-19 and a regained sense of control to return to normalcy" (Miao et al. 2022, p. 2).

Countries where tourism had a great weight in their economies were particularly affected by the pandemic crisis, as is the case with Portugal. Tourism is Portugal's main

economic export activity (51.1% of service exports) and contributes to 8.0% of national GDP, which is significantly above the average of OECD countries, where the sector contributes directly to 4.4% of GDP and 21.5% of service exports (OECD 2020). Furthermore, Portugal is highly reliant on inbound tourism, representing 69% of total tourism expenditure, while domestic tourism (31%), which has been less affected by this crisis, is significantly lower than the average of OECD countries (75% of the total tourism economy) (OECD 2020). In the national context, there are large regional differences, with the Algarve and Lisbon concentrating more than half of the total overnight stays in the country (55.5% in 2019), compared to the scarce 4.3% of the Azores.

However, tourism has become one of the main development pillars in this archipelago. In 2015, tourism contributed to 14.1% of the GDP in the Azores and was responsible for more than 10% of employment (SREA 2018). Currently, these values are even more expressive, as noted in the increase of 103.3% of nights in tourist accommodation between 2014 and 2019, which corresponded to more than double the national average (Table 1).

**Table 1.** Tourism activity in the Azores and Portugal in 2019 (INE 2022; SREA 2022).

|  | Overnight Stays | Guests | SR [1] | AS [2] | FG [3] | GPI [4] | GPS [5] | OV [6] |
|---|---|---|---|---|---|---|---|---|
| **Portugal** | 77,882,700 | 29,495,400 | 36.3% | 2.6 | 60.5% | 286.6 | 319.9 | 44.0% |
| **Azores** | 3,009,845 | 971,794 | 39.9% | 3.0 | 49.6% | 400.3 | 418.5 | 103.3% |

[1] SR—Seasonality rate; [2] AS—Average stay; [3] FG—Foreign guests; [4] GPI—Guests per inhabitant × 100; [5] GPS—Guests per sq. km; [6] OV—Overnight variation 2014–2019.

In addition, the number of visitors, when weighted with the surface of the territory (guests per sq. km), the number of inhabitants (guests per inhabitant × 100) and the size of its economy, is significantly higher than the Portuguese average (Table 1).

The importance of tourism for this autonomous region is further accentuated by its geography and economy. This small archipelago, with scarce 2346 km$^2$ and approximately 240 thousand inhabitants, is an ultraperipheral territory, with nine islands scattered in the North Atlantic Ocean, and limited accessibilities, which are factors that restrict the competitiveness of its economy (European Commission 2019).

Its natural and cultural resources, its uniqueness, and the association of the islands with exotic and authentic areas, have led to the appreciation of tourism in the region. The destination's brand image is related to nature-based tourism and sustainability, so activities such as whale watching, hiking, and canyoning, arise as strategic tourism products.

However, the COVID-19 pandemic has drastically interrupted the growth cycle of tourism in these islands. The answer of the Azores authorities to the pandemic situation was swift. On 3 March, even before the first detected positive case in the region (15 March), the Regional Government declared a state of contingency. In general, the number of cases in the region was kept at a relatively low level, which led to its inclusion on the list of safer destinations during the summer and, therefore, fewer travel restrictions imposed by several important international tourism markets, namely the United Kingdom and other European countries (Safe Communities Portugal 2020). At the initial stage, the combination of measures adopted by other countries, by the Portuguese government, and by the Regional Government, resulted in the closure of international traffic to the region. Flights only restarted later, with several restrictions for incoming travelers.

In the following months, the Regional Government's strategy to stimulate tourism demand had tight travel restrictions, based on two pillars. The first was to reduce the likelihood of virus transmission from abroad by imposing control measures on inbound travellers, including the obligation to submit a negative COVID-19 test upon arrival, and a second test six days later, for anyone staying in the region for at least seven days. The second was to stimulate inter-island regional tourism by launching the «Viver os Açores» programme, intended to "encourage holiday and leisure trips in the Azores through its islands, as well as to promote the economic recovery of the region through tourism" (Presidência do Governo 2020, p. 1). According to a study conducted by the Tourism

Observatory of the Azores (OTA 2020), this campaign resulted in almost 75% of Azoreans changing their travel plans due to the COVID-19 pandemic, and 84.2% of those who still intended to take a holiday to choose the Azores. Despite these numbers, the global effects of the campaign on tourism were rather limited, and the Azores ended up being the Portuguese region that in 2020 had the biggest reduction in the number of overnight stays (−71.9%) (INE 2022). This reduction was more relevant on islands with greater tourist demand than on smaller islands (SREA 2022). An important recovery was noted in 2021, with the number of overnight stays more than doubling (114.2%) although still significantly lower than in 2019 (45.1% lower). In 2022, and still with some restrictions on travel, tourism demand in the archipelago finally managed to recover its pre-pandemic results.

The Azores are the Portuguese territory where the impact of the COVID-19 pandemic on tourism was most significant. Significant explanatory factors include insularity (dependency on air transport) and domestic market limitations (Castanho et al. 2021; European Commission 2022). However, it is important to understand the impacts on the various tourism clusters and the specific differences within these subsectors. Since nature and active tourism are the core products and the Azores brand image (IPDT 2016), it was decided to analyse the canyoning provider companies, as it is an emerging product with significant development potential. Thus, this study analyses the tourism recreation subsector, seeking to understand the effect of the pandemic on these companies, and which adaptations were made in their business models.

A mixed methodological approach was followed, using questionnaires and interviews as empirical instruments to collect information: the first during the most impactful period of the pandemic, and the second already in a period of tourism recovery. As a niche of companies in a limited territory, it is essential to simultaneously compare them with other recreation products in the Azores and nationwide. The investigation allowed the authors to understand the resilience of these companies and the changes driven by this intense and unexpected crisis.

## 2. Canyoning in the Azores

Canyoning is both an adventure sport and a leisure activity, consisting of the progressive exploration of mountain streams using a variety of techniques, such as abseiling, scrambling, jumping, and swimming (Silva et al. 2020). The enjoyment of natural beauty and the challenges and fun associated with abseiling down waterfalls and jumping into natural pools are some of the relevant elements of this activity (Hardiman and Burgin 2011).

This activity can either be practised autonomously or provided by business technicians, namely, within the scope of the services offered by tourism recreation companies. Thus, as far as this product is concerned, the market can be divided into two broad groups: (i) canyoning enthusiasts, with the technical knowledge to carry out the activity autonomously and who will travel with the main purpose of practising canyoning; and (ii) a highly diverse range of travellers, with little or no knowledge of the sport, but who want to enjoy engaging experiences with strong discovery, adventure and fun components (Silva et al. 2020). This second group is the main market for tourism recreation companies specializing in canyoning.

The Azores boast the highest supply of canyoning routes in Portugal, with 114 itineraries spread across six of the nine islands in the archipelago: Flores (44); São Jorge (31); São Miguel (21); Santa Maria (8); Faial (8); and Terceira (3). This is a relatively new product in the region, but it has quickly emerged as one of the most important strategic products in some of these islands (IPDT 2016; Silva et al. 2020).

Although the islands of Flores and São Jorge have the best natural resources for the practice of this sport, it is in São Miguel that canyoning supply has greater economic relevance. The reason for this apparent contradiction is justified by the clear concentration of tourists visiting the Azores on the island of São Miguel (64.2% of total guests in hotel accommodations in 2019) (SREA 2022), in addition to the use of several simple and sportive

canyoning routes. Consequently, the number of service providers increased in São Miguel, which currently holds 42% of the total number of canyoning companies in the region.

Presently, there are 12 canyoning companies offering canyoning in the Azores, spread across six islands: five in São Miguel, two on the islands of Flores and São Jorge and one in Santa Maria, Faial, and Terceira. Of these 12 companies, 11 are microenterprises, and canyoning is their main product. According to questionnaires and interview results, the weight of canyoning in these companies' total supply varies between 40% to 59% (three companies); 60% to 79% (three companies), and for five of them, it represents over 79%. Only in one of these companies (medium-sized) canyoning is not the main product; however, it still represents between 20% and 39% of the company's turnover.

Although canyoning is a seasonal product, with a high concentration of services provided between July and September (the weight of over 40% in all these companies and between 60% to 79% in most of them), all of the businesses provide canyoning services throughout the year. In Azores, as in most other regions, canyoning services tend to target visitors who wish to enjoy a one-off activity with a relatively low level of expertise and for a short period. Thus, the predominant service model in the Azores consists of offering a "baptism" or "discovery" activity, with the following characteristics:

- Duration—approximately half a day, 3 to 4 h of activity;
- Level of difficulty—easy routes, with rappel descents of below 20 m;
- Groups: small groups of 1 to 10 persons accompanied by 1 to 2 monitors;
- Requirements—being able to swim, to walk on uneven ground and having a minimum age of 5 to 14 years, which varies according to the company;
- Price—between 50 and 70 euros;
- Includes—wetsuit, footwear, personal protective equipment, insurance, technical framework;
- Complementary or other characteristics—transport, snacks, photos.

Some companies also offer services to experienced customers, namely more technical routes, and hybrid products, combining hiking, gastronomy, and canyoning, or canyoning and coasteering.

## 3. Methodology

In addition to researching sources with statistical data and information on the evolution of tourist activity in the Azores and on the impact of the COVID-19 Pandemic, an empirical study was carried out, supported by a mixed approach with both quantitative and qualitative data.

In the first phase, questionnaires were applied between October and November 2020 to tourism recreation companies in Portugal, focusing on the impacts of the COVID-19 pandemic on their businesses, their levels of resilience, and answers to this crisis. Despite the focus of this study being specialized canyoning businesses in the Azores region, the option of considering all recreation companies in Portugal as a universe allowed the authors to ensure the representativeness of the sample and to understand if there were regional differences between activities. As the legal registration of these companies is mandatory in Portugal and available in the National Tourism Registry database (n = 9575), the email addresses of all registered companies were used to invite business owners to answer this questionnaire, available on an online platform (Turismo de Portugal 2020b).

Across Portugal, these companies sell a huge diversity of activities, which may be divided into three broad groups: (i) cultural tourism and landscape touring; (ii) outdoor/nature tourism and adventure tourism; and (iii) nautical operators. Approximately 99% of these companies are micro or small companies that heavily rely on foreign market customers (65%), providing mainly half-day or one-day services (66% of the offered activities) (Turismo de Portugal 2020a).

The application of the survey aimed to analyse the effects of the COVID-19 pandemic on tourism recreation companies in Portugal, the relevance of the adopted measures to mitigate the effects of this pandemic and the resilience of the sector. It was also intended

that these results could contribute to the construction of the interview script to be applied to canyoning companies in the Azores, and to establish a comparison of these companies with the recreation sector average. The questionnaire considered 32 questions grouped into four dimensions: (i) the characterization of the company (n = 10); (ii) the impact of the pandemic on companies (n = 12); (iii) the analysis and evaluation of measures to support the sector (n = 3); and (iv) transformation, resilience, and trends in these businesses (n = 7).

A total of 419 validated answers was obtained, corresponding to a response rate of 4.5% and a margin of error of 4.67% for a 95% confidence interval. In terms of tourism recreation companies operating in the Azores, 43 responses were considered valid, corresponding to 10.3% of the study's sample.

In the second phase, between February and March 2022, semi-structured interviews were carried out with all the tourism recreation companies specializing in canyoning in the Azores (n = 12). The type of assessment tool chosen and the period in which it was implemented allowed a detailed and qualitative collection of information once companies were returning to pre-pandemic demand levels. These interviews aimed to analyse the impacts of the COVID-19 pandemic on their businesses, the evaluation of support measures made available to companies, how they adapted to the imposed restrictions, what changes there were in their supply chain and in their business model, and what the prospects for the future were.

The main questions in the interviews were organised in five dimensions: (i) the characterization of the respondent and the company; (ii) canyoning product, supply, demand, and markets; (iii) the assessment of support measures during the pandemic period; (iv) impact, resilience, adaptations, and changes resulting from the pandemic; and (v) expectations regarding the future.

The interviews had an average duration of 57 min, ranging from 00:27 to 1:27 min, and were carried out remotely and recorded using the Zoom platform. Eleven respondents were the companies' CEO, and the one left was the CFO responsible for the canyoning product.

## 4. Results—Impact and Answers to the COVID-19 Pandemic on the Azores' Canyoning Services

Data obtained from the questionnaires confirmed other studies' information. This is a sector in which most companies are very small (59.2% of companies have less than three full-time workers), operate throughout the year (91%), and cover a wide range of activities, highlighting cultural circuits, nature tourism, nautical activities, and events organization.

In the specific case of the 12 canyoning companies in the Azores, three did not have any full-time workers, five had between one and two, and only one had more than 20 full-time workers. The weight of canyoning in their businesses in 2019 was over 75% for seven companies and between 30% and 50% for the rest. Only in two of these companies, canyoning was not their main product, however, it still represents between 20% and 39% of the company's revenue. Although this is a seasonal product, with the highest concentration of services provided between July and September (with a weight of over 40% in all companies and 60% to 79% in most of them), all the companies provide services throughout the year.

The impact of the pandemic on Portuguese tourism recreation companies was very evident in the first year. The reduction in these businesses in the second quarter of 2020 was wide, with most agents having to close their activity. Regarding the third quarter, which corresponds to the sub-sector's peak season, 98% of the companies reported a decline in their activity, and in 35% of them, this decrease was above 79% (n = 43). In the national panorama (n = 383), results were also striking, though slightly less expressive, as 89% of the companies considered a turnover reduction, of which 30% assumed a decrease of over 79%. It should be noted that some companies decided to temporarily close their business in 2020, as was the case of one of the 12 canyoning companies in the Azores region.

Comparatively, the reduction in specialized canyoning companies in the Azores was relatively lower, particularly in those that suffered a reduction of 40% to 59% (four compa-

nies) or above 79% (three companies), but slightly higher than the average for companies with canyoning services in Portugal (Table 2). All companies with a reduction of over 79% operate on São Miguel Island. These figures demonstrate not only the huge negative impact of this pandemic on these businesses but also the great heterogeneity among the companies.

**Table 2.** Turnover variations in the third quarter of 2020.

| Company Group | Increased | Maintained | Reduced | Reduced | | | | |
|---|---|---|---|---|---|---|---|---|
| | | | | <20% | 20–39% | 40–59% | 60–79% | >79% |
| Portuguese recreation companies (n = 419) | 4% | 7% | 89% | 26% | 12% | 17% | 16% | 30% |
| Portuguese canyoning companies (n = 38) | 3% | 5% | 92% | 23% | 11% | 29% | 17% | 20% |
| Azores recreation companies (n = 43) | 2% | 0% | 98% | 28% | 14% | 12% | 12% | 35% |
| Azores canyoning companies (n = 12) | 0% | 0% | 100% | 9% | 18% | 36% | 9% | 27% |

These differences were not statistically significant, according to the Mann-Whitney test. The option for this non-parametric test is justified as the variables are qualitative and do not present a normal distribution (MacFarland and Yates 2016). Results indicate that there are no significant differences between the specialized canyoning companies in the Azores and the total number of tourism recreation companies in Portugal in terms of their turnover reduction during the third quarter of 2020 (U = 1882.00, $p$ = 0.351), nor in the comparison between the total number of recreation companies in the Azores and the national ones (U = 7389.00, $p$ = 0.943).

Regarding market deviations, three classes of analysis were established: (i) local community; (ii) tourists from other Portuguese regions (national tourists); and (iii) international tourists. Table 3 depicts the variations in these markets during the peak high season quarter. A sharp reduction may be observed in the international market and a less noticeable decrease in the domestic market. This observation also reveals high heterogeneity in the sector and a difference between canyoning companies, compared to the general recreation sub-sector. All canyoning businesses reported an increase in demand from the local market, with more than half reporting an increase from the domestic market, which was not the case for most tourism recreation businesses.

**Table 3.** Market variation in the third quarter of 2020.

| Company Group | Markets | Increased | Increased | Maintained | Reduced | Reduced | Reduced |
|---|---|---|---|---|---|---|---|
| | | 26–50% | 1–25% | | 1–25% | 26–50% | >50% |
| Azores canyoning companies (n = 11) | Local community | 64% | 36% | - | - | - | - |
| | National tourists | - | 55% | 9% | 9% | 18% | 9% |
| | International tourists | - | - | - | - | 36% | 64% |
| Azores recreation companies (n = 43) | Local community | 17% | 19% | 21% | - | 5% | 38% |
| | National tourists | 10% | 14% | 17% | 10% | 5% | 45% |
| | International tourists | - | - | 2% | 5% | 19% | 74% |
| Portuguese recreation companies (n = 419) | Local community | 4% | 11% | 31% | 5% | 5% | 43% |
| | National tourists | 11% | 11% | 19% | 7% | 6% | 45% |
| | International tourists | 1% | 1% | 4% | 5% | 7% | 82% |

Azores canyoning companies' managers were unanimous in stating that the main cause of the reduction in their turnover was the sharp fall in the number of overseas visitors. When confronted with the level of safety of their activities in relation to virus transmission, they all claimed to have taken the appropriate measures to ensure the safe practice of the sport.

Regarding the impact of this pandemic on employment, the majority of canyoning companies and recreation services in the Azores stated that they did not intend to dismiss employees or cancel contract renewals (67% and 70%, respectively). These are relatively high figures and point to the possibility of a major negative impact on employment in the region (Table 4). As for the national scenario, the data also revealed some uncertainty with respect to these possibilities, with 19% of companies answering: "I don't know, perhaps".

**Table 4.** Companies that consider dismissing or not renewing contracts.

| Azores Canyoning Companies (n = 12) | | | Azores Recreation Companies (n = 43) | | | Portuguese Recreation Companies (n = 419) | | |
|---|---|---|---|---|---|---|---|---|
| Yes | Perhaps | No | Yes | Perhaps | No | Yes | Perhaps | No |
| 25% | 8% | 67% | 23% | 7% | 70% | 24% | 19% | 57% |

According to Azores' canyoning companies, the main factors that positively influenced the business during this COVID-19 pandemic were: (i) the successful adaptation of their products (73%); (ii) the possibility of maintaining the company's regular service (73%); (iii) the successful adaptation of the company's structure to accommodate the demand (64%); and (iv) the availability of public support measures (55%). Regarding the negative factors, the entrepreneurs highlighted: (i) travel limitations among the affected countries (82%); (ii) the applicable rules in terms of social confinement (73%); and (iii) a significant drop in tourism demand (55%).

Within the scope of the governmental measures, one of the most emblematic initiatives was the implementation of the Clean and Safe seal. All canyoning companies in the region have implemented this measure (except the company that closed in 2020), while the proportion of Azores' recreation companies (n = 43) to implement this measure was 72%, and the total Portuguese recreation companies (n = 338) was 82%. When asked to evaluate national and regional government support measures, the respondents highlighted the suspension of taxes and contributory payments and the implementation of the Clean and Safe seal as the most relevant actions. As far as the "simplified layoff" measure and its extension to managing partners is concerned, around half of the companies considered this to be very important, while the other half considered it unimportant. The least valued measures were the possibility of a moratorium on the payment of interest and capital on existing loans, and access to new loans with subsidized interest or state guarantees.

As for the level of satisfaction with the actions of the sub-sector's responsible entities, the companies evaluated them mainly with the following answers: "neither satisfied nor dissatisfied" or "no opinion (indifferent)". The level of dissatisfaction was higher in relation to the national government and public entities when compared to the regional reality. The main difference between the assessment by canyoning companies and the rest of the Azores' recreation businesses was the level of satisfaction with municipalities' support measures, with higher dissatisfaction on the part of the recreation companies in general (Table 5).

**Table 5.** Satisfaction with public entities' measures.

| Company Group | Public Entities | Unsuitable | Indifferent | Adequate |
|---|---|---|---|---|
| **Azores canyoning companies (n = 12)** | Government and public entities—mainland Portugal | 36% | 46% | 18% |
| | Government and public entities—Azores | 18% | 64% | 18% |
| | Municipalities | 9% | 64% | 27% |
| **Azores recreation companies (n = 43)** | Government and public entities—mainland Portugal | 33% | 42% | 26% |
| | Government and public entities—Azores | 21% | 47% | 33% |
| | Municipalities | 36% | 46% | 18% |

As the interviews were carried out in 2022, restrictions were much lower and expectations were brighter in the short term, thus allowing the authors to have a perspective of what the post-pandemic period would be like. This viewpoint also allowed the investigators to analyse resilience and transformations in the business models of these companies. Of the 12 canyoning companies in the Azores, only one was temporarily closed between 2020 and 2021, and therefore it is not considered in some of the presented data. The strong reduction in businesses in 2020 was confirmed, with an average decrease of 65.3% in the number of canyoning customers, ranging between −18.1% up to −90.6%. However, in 2021, there was a strong recovery with an average growth of 149.0% compared to the previous year, showing a reduction of 13.5% when compared to 2019 values. As can be seen in Table 6, some heterogeneity may be found between companies, due to several factors, namely the markets, business size, corporate structure and business models, geographic location, and the implemented answers.

**Table 6.** Variation in the number of canyoning customers in the Azores.

| | Reduction (%) | | | | Growth (%) | | | | |
|---|---|---|---|---|---|---|---|---|---|
| | **Over 75** | **51–75** | **26–50** | **1–25** | **1–50** | **51–100** | **101–200** | **201–300** | **Over 300** |
| **2019–2020** | 4 | 2 | 3 | 1 | 1 | 0 | 0 | 0 | 0 |
| **2020–2021** | 0 | 0 | 0 | 1 | 0 | 3 | 4 | 2 | 1 |
| **2019–2021** | 0 | 2 | 2 | 2 | 3 | 1 | 0 | 1 | 0 |

For five of the 11 companies, demand in 2021 was even higher than in 2019, when the number of customers was around 12,500 and the estimated business value was 750 thousand euros. These values are relatively low, but the product is already considered strategic for the destination (IPDT 2016) and has strong growth potential. Four main factors contribute to this perception: (i) in 2019 only about 1.3% of tourists who visited the region have performed this activity; (ii) this product is capable of providing experiences of great value (Botelho et al. 2022; Hardiman and Burgin 2011); (iii) it has a broad market (families, youth to adults, without the need for technical knowledge); and (iv) the Azores have excellent natural resources for carrying out this activity (Silva et al. 2014).

However, the pandemic brought significant market changes; by 2020, it was almost entirely local and national. These markets remained important in 2021, but there was a strong recovery of the international market, and it is foreseeable that in 2022 this will be the predominant market for all companies.

As most of these companies are microenterprises, and in 2019 four of them did not have any permanent workers, along with canyoning's seasonality, several companies use temporary or occasional guides. This HR structure contributed to greater resilience in the sector, but also limited access to some support measures. Five of the companies did not receive any support during the crisis, either because they did not apply for it or because they were not eligible. For three of the companies, this support was not very expressive, and for the remaining four, it was fundamental, particularly for companies operating in São Miguel and those with a greater number of permanent workers.

Business responses to the crisis were diverse. One of the companies closed its activity until mid-2022, and those that did not have permanent human resources survived, especially because the company's CEO had another professional activity and because operating costs were low. However, most sought to adapt their business model and develop other actions that would mitigate the impacts resulting from the demand reduction. A significant number of companies redirected their supply to other markets and even changed products, focusing on local community customers, and creating holiday camps. Three of the companies extended their business to other activities, such as tree climbing and vertical work.

Regarding the adaptation of canyoning services, these were essentially related to the reinforcement of equipment hygiene, adherence to the national Clean and Safe seal, and implementing measures such as distancing and mask-wearing. However, because the

activity is performed outdoors and includes a significant physical and aquatic component, some of these procedures were not followed. Several companies adopted other measures such as reducing group sizes and five companies stopped transferring customers between the accommodation and the activity locations. This last procedure allowed them to reduce costs and facilitate the management of the activity, which is why they consider it a measure to keep, especially since most customers are tourists who resort to car rentals during their holidays.

The resilience of these companies is immediately evident as they all survived the crisis and did not fire any employees, although they have significantly reduced the hiring of temporary workers. Some entrepreneurs even mentioned that the crisis also had positive impacts, highlighting that they had time to invest in business model planning, promotion and digital marketing, managing equipment, diversifying products and markets, and improving product management.

Regarding incorporated measures that businesses intend to keep, most say that they are not significant, only mentioning suit hygiene and the end of customer transfer. Regarding their business model, there was a positive evolution with four companies that did not have any permanent employees, now having and reinforcing the acquisition of equipment. In terms of structuring changes, the strengthening of the local market (clients of canyoning and holiday camps) should also be highlighted, which helps to reduce the seasonality of the activity and to focus on other business segments. Two companies started tourist accommodation businesses, three added vertical work and tree climbing segments and the other three started other recreation activities such as coasteering, holiday camps, and adventure parks, among others.

Although some companies believe that product diversification is an important strategy for increasing their company's competitiveness and sustainability, the majority prefer to continue focusing primarily on the canyoning product because they believe it is profitable, has strong growth potential, and that their specialization in this product allows them to strengthen their competitiveness.

## 5. Conclusions

Tourism was one of the main sectors to be affected by the COVID-19 pandemic crisis (OECD 2020; UNWTO 2022). To mitigate the effects of this crisis, efforts were redirected to domestic markets and the adoption of sanitary measures to promote safety and stimulate tourists' confidence to visit destinations. Countries such as Portugal, or the Azores region, heavily dependent on foreign tourism, had greater difficulties mitigating the impact of this crisis, both economically and socially. However, tourism's strong recovery in 2022 contributed significantly to the high performance of Portuguese GDP, which will be around 6.7% in 2022 (Banco de Portugal 2022).

The Azores managed to ensure a relatively low level of COVID-19 risk until early November 2020. Although some domestic tourism-oriented measures were implemented in the region, results still demonstrate severe reductions in 2020, when compared to the average of OECD countries and the Portuguese reality (European Commission 2022; INE 2022). These impacts were more expressive in islands with a higher tourist weight, namely Faial and São Miguel, which may be explained by the region's dependence on external tourism and accessibility limitations, as the aeroplane and cruises are practically the only means to visit the archipelago. The progressive reduction of travel restrictions allowed the recovery of tourism results, which started in 2021 and was accentuated in 2022 (SREA 2022).

As for the impact of the pandemic on the tourism recreation sub-sector in Portugal and the Azores region, 2020 was a dramatic year. However, most companies have survived this crisis and are relatively optimistic about the future. This position may be justified by the fact that most of these companies are microenterprises and have relatively low fixed costs. Therefore, the recreation sub-sector may be considered one of the most resilient in the tourism sector.

Data collected through questionnaires showed that during the period with the greatest impact on business, there were no statistically significant differences between the reduction in tourism recreation companies in Portugal and those in the Azores, including the ones specializing in canyoning in that region.

Regarding the interviews carried out with the Azores canyoning companies in 2022, at a time of strong recovery of their activity, it was confirmed that all these companies survived this pandemic crisis. Answers differed between companies depending on their business model, size, location, and the level of expertise. All these agents highlighted that the crisis had strong negative impacts on their businesses, but also identified advantages such as the "discovery" of the local market and an opportunity to dedicate time to planning their businesses. In general, companies took the opportunity to accelerate some of the transitions already underway or to promote changes in their business model, with a focus on resilience and competitiveness—to answer a demand that they believe will be growing in the coming years.

There are several studies on the impacts of the COVID-19 pandemic on the tourism sector and these businesses' resilience over adversities caused by this crisis, but few analyse companies specialized in a product and/or a region. In this sense, this study contributes to reinforcing the conclusions of macro-level studies and simultaneously understanding a set of territorial characteristics of canyoning as a specific tourism product.

However, as the interviews were only carried out with canyoning companies in the Azores, the generalization and comparison that was established with the set of tourism recreation companies in Portugal, and even canyoning companies in Portugal, should be considered limited and could be valued in the future with the application of interviews to a broader sample of companies.

**Author Contributions:** Conceptualization, F.S. and T.L.; methodology, F.S., T.L. and M.S.; validation, F.S., T.L. and M.S.; investigation, F.S. and T.L.; writing—original draft preparation, F.S., T.L. and M.S.; writing—review and editing, F.S., T.L. and M.S.; supervision, F.S.; project administration, F.S. All authors have read and agreed to the published version of the manuscript.

**Funding:** This research was funded by Portuguese national funds through the FCT—Fundação para a Ciencia e Tecnologia, I.P., under the Project grants UIDB/00295/2020 and UIDP/00295/2020.

**Institutional Review Board Statement:** Not applicable.

**Informed Consent Statement:** Informed consent was obtained from all subjects involved in the study.

**Data Availability Statement:** Not applicable.

**Acknowledgments:** The authors of this article thank all the tourism recreation companies who answered the questionnaire, and, particularly, those responsible for Azores canyoning who consented and allowed the interviews to be carried out.

**Conflicts of Interest:** The authors declare no conflict of interest.

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
