# Peer review of "The Resilience of Tourism Recreation Companies in a Pandemic Context: The Case of Canyoning in the Azores"

_socsci, doi:10.3390/socsci11120558_

Round 1
Reviewer 1 Report
Comments and Suggestions for Editor and Authors
The question of "the level of crisis caused by the Covid-19 and the resilience of tourism recreation companies specialized in canyoning in the Azores region and whether these reactions led to more sustainable business models" is a relevant and important topic. It is very important for the academic community to develop research on these current issues, which greatly contribute to sustainability. It is very important for the academic community to develop research on these current issues, which greatly contribute to sustainability. Tourism in the Ultraperipheral regions in Europe must be increased to allow for a better European balance, through the development of attractive peripheral regions. The abstract is well written and well dimensioned. The introduction and literature review are written in a thoughtful and balanced way. The methodology and the discussion of the results are presented in a clear and objective way. The conclusions are presented in a clear and objective way. The bibliography is extensive and current.
Other issues related to the work presented:
1. What is the main question addressed by the research? Is it relevant and interesting?
The work aims to identify "the level of crisis caused by the Covid-19 and the resilience of tourism recreation companies specialized in canyoning in the Azores region and whether these reactions led to more sustainable business models". The research that was carried out is oriented to fulfill this proposed objective. It is a very relevant, interesting, and current topic, not only for the academic world, but for global society. One of the major contributions to the GDP of the Azores islands comes from tourism. This is a relevant, pertinent, and interesting question as a current research topic.
2. How original is the topic? What does it add to the subject area compared with other published material?
The theme is original and relevant for sustainable development in the European Ultraperipheral regions. The way the authors approach the issue, using scientific methodology, and having the work well structured, is an important contribution to knowledge. It is a paper with quantitative and qualitative methodology, which fully demonstrates the results of the study. This is a successful work in this area of investigation.
3. Is the paper well written? Is the text clear and easy to read?
Yes, the text is well written, in a clear, objective, and precise way.
4. Are the conclusions consistent with the evidence and arguments presented? Do they address the main question posed?
The conclusion presents the main aspects discussed in the work. It is written in an objective, simple and very understandable way, but it can be improved. The conclusion focuses on the main observations of the results obtained by the methodology used. The conclusion is balanced and parsimonious, however it should present the limitations of the work, and open perspectives for future work that may eventually be carried out, based on this paper. Therefore, it remains to add to the conclusions the limitations of this work, and what impact this work hopes to have (academic environment and society in general).
5. Is there any current bibliography that can be suggested, with the aim of improving the paper?
No, but the final bibliography must be standardized, according to the requirements of the Journal.
Round 2
Reviewer 2 Report
I am still unable to tell what is the research question or the contribution of the paper after reading the introduction.
Author Response
Dear reviewer
Thanks for the review.
To respond to the suggestion presented, we added two paragraphs at the end of the Introduction and we improved the English.
Best regards